# High-Protein Processed Foods: Impact on Diet, Nutritional Status, and Possible Effects on Health

**DOI:** 10.3390/nu16111697

**Published:** 2024-05-30

**Authors:** Rosa M. Ortega, Nerea Arribas-López, María Dolores Salas-González, Aránzazu Aparicio, Liliana Guadalupe González-Rodríguez, Laura M. Bermejo, María Del Carmen Lozano-Estevan, Esther Cuadrado-Soto, Ana M. López-Sobaler, Viviana Loria-Kohen

**Affiliations:** 1Department of Nutrition and Food Science, Faculty of Pharmacy, Complutense University of Madrid, 28040 Madrid, Spain; rortega@ucm.es (R.M.O.); nerearri@ucm.es (N.A.-L.); masala06@ucm.es (M.D.S.-G.); araparic@ucm.es (A.A.); liligonz@ucm.es (L.G.G.-R.); mlbermej@ucm.es (L.M.B.); mlozan16@ucm.es (M.D.C.L.-E.); esther.cuadrado@ucm.es (E.C.-S.); asobaler@ucm.es (A.M.L.-S.); 2VALORNUT Research Group, Department of Nutrition and Food Science, Faculty of Pharmacy, Complutense University of Madrid, 28040 Madrid, Spain; 3San Carlos Health Research Institute (IdISSC), 28040 Madrid, Spain

**Keywords:** high-protein processed foods, health, marketing trends, overconsumption

## Abstract

Proteins are macronutrients with multiple health benefits, but excessive consumption can negatively affect health. This study aimed to evaluate the characteristics of a sample of high-protein processed foods (HPPFs), describe how their consumption affects dietary balance, and acquire knowledge of the consumption patterns of these products in a Spanish population. A sample of HPPFs available in supermarkets and on websites was collected. The contribution to recommended protein intakes was calculated using national and international references and considering the single consumption of the HPPFs and the product plus 150 g of meat. Furthermore, an online survey was conducted among a convenience sample. A total of 36 enriched protein products were evaluated. The percentage of proteins in these products ranges from 10 to 88%. The contribution of the protein recommended intake was within a range of 87.4–306.6% and 66.4–232.8% (women and men, respectively), only considering the additional proteins from 150 g of meat. One hundred thirty-nine participants completed the survey; 67.6% affirmed that they had consumed HPPFs, and half consumed them without following any consumption control. Since these products are accessible to everyone in supermarkets and protein intake is generally higher than the recommended limits, regulating the mass sale of HPPFs is essential to ensure they do not lead to protein overconsumption.

## 1. Introduction

Nowadays, society is constantly changing, and changing dietary habits is part of that [1,2]. Among these changes, a decrease in cereals, potatoes, and legume consumption, which leads to a significant reduction in the percentage of energy from carbohydrates in the diet, and, on the opposite, an increase in the consumption of meat, with the consequent increase in lipid and protein intake, have been described [2]. 

Proteins are a fundamental part of cells and essential for tissue growth, repair, and renewal [3,4,5,6]. Adequate protein consumption provides the amino acids needed to function and adequately maintain vital organs and immune cells [7,8]. Similarly, proteins act as regulators and transporters at the molecular level [8], also playing a crucial role in satiety [9,10]. That is why the consumption of protein in our diet is essential. 

However, although proteins have essential health effects, their excessive consumption could have adverse effects and increase this risk in the long run as consumed proteins rise, with several systems being affected. 

In this sense, several studies have shown that excessive consumption of proteins produces an increase in the volume and weight of the kidney, acting as a physiological modulator and also prompting a temporary increase in glomerular filtration, which is one of the factors in the fast progression of kidney disease or kidney failure [3,10,11,12,13,14]. Similarly, when diets with a high protein content are adhered to, the amount of urea in the blood rises, leading to renal overload, which could cause kidney function loss in the long run [3,11,14].

Some studies have suggested an increase in the risk of hypertension linked to high protein consumption. This hypothesis is based on the idea that when stimulated by a high protein load, immune cells may release free radicals, cytokines, and other vasoactive factors, promoting increased blood pressure. However, it is known that certain amino acids could have a protective effect on blood pressure [15].

In addition, sulfur amino acids (mostly coming from animal proteins) may result in physiological acidity that could affect the health of bones in the future, as well as being related to processes of hypercalciuria and hypocitraturia, risk factors for the development of nephrolithiasis or kidney stones composed of calcium oxalate [7].

Another undesirable effect of excessive protein consumption is a rise in the risk of prediabetes and type 2 diabetes. This consumption adversely affects insulin action, increasing this risk by 20–40% for every 10 g of protein ingested over 64 g [8]. Additionally, an excess in protein consumption without lowering fats or carbohydrates could contribute to an increase in energy intake and, therefore, the development of obesity [8].

Finally, regarding gut health, excessive protein consumption could increase the transfer of nitrogen compounds in the large intestine, potentially altering the gut microbiota’s composition and diversity. It could result in modifying its metabolic activity and causing issues in the production of bacterial metabolites, which would have repercussions on the metabolism, physiology, and health of the mucosa of this portion of the intestine [10].

According to the ANIBES study (by its acronym in Spanish, “*Estudio de Antropometría, Ingesta y Balance Energético en España*”), protein intake in the Spanish population rounds 16.8% of the total energy intake. This percentage is above the European Food Safety Authority (EFSA) recommendation, marked at 15%. The ANIBES study showed a high contribution of proteins to daily energy intake, which increased with age [1]. This matter was highlighted in the analysis of macronutrient availability data between 1964 and 2011 in Spain. According to these data, the evolution of Spanish households’ diets showed a reduction in the contribution from complex carbohydrates, followed by an increase in proteins [16]. It was also identified in other national studies, such as ENALIA and EsNuPI, conducted on the Spanish pediatric population [17,18].

Additionally, due to these worrying data that show an unbalanced diet because of excess protein consumption, consumers are increasingly looking for foods rich in protein because this diet is popularly associated with health and increased muscle mass [19,20].

As a result of the increase in demand and the fashion of consuming these products, the market has started to offer several protein-enriched foods. Consequently, its market has grown considerably in the past few years [19,20]. All these products that used to be available in specialized stores, generally associated with sports practice, started to be available in regular food markets, reaching the general population [20].

The increase in protein consumption can cause an imbalance in our diet, with possible negative repercussions on health. This situation may be more alarming within the context of a population that, due to its eating habits, already consumes more than the recommended protein intake. To date, we ignore the effect that consuming a massive supply of high-protein processed foods can have on the total protein intake or the frequency and reasons consumers buy these products. 

Considering all these facts, the objective set out for this research was to analyze a sample of high-protein processed foods (HPPFs) available in different supermarkets and evaluate how their consumption can increase the percentage covered by the recommended intake of that nutrient. Moreover, the other objective was to evaluate the rate at which people buy these products and the reasoning behind their choices. The results of this study will allow us to determine to what extent its consumption could affect the balance of the diet and guide health professionals in their recommendations.

## 2. Materials and Methods

To carry out the analysis of HPPF products, different food items were compiled from the main physical supermarket chains in the Community of Madrid, Spain (Mercadona, Lidl, Carrefour, and Ahorramas) and online stores (Yopro, Foodspring, and Prozis) from February to May 2023.

The inclusion criteria for foods were as follows: (1) being sold in the Spanish market and (2) processed foods that included nutritional claims as “source of protein” (at least 12% of the energy value of the food is provided by protein) and “high protein” (at least 20% of the energy value of the food is provided by protein). The products were grouped into dairy products, jellies, energy bars, snacks, breakfast cereals, breads, and creams, and a more significant number of products were selected from those categories that had a broader product offering (it was especially dairy products).

The nutritional labels of these products were used to compile the following data: grams of proteins provided per 100 g of product, the total energy (kilocalories) by container or serving, the type of added protein, the target group of people, and the nutritional statements or claims present on each product. Information on the protein content of a similar non-enriched product (NEP) was compiled. Data from food items were collected in an Excel database specifically designed for this work. Subsequently, the contribution to recommended protein intakes was calculated using national references [Recommended Daily Intakes (RDIs) by Ortega et al., 2019] [21] and Moreiras et al., 2016 [22], and international references [Population Reference Intake (PRI) by The European Food Safety Authority-EFSA] [23]. This contribution was calculated considering the exclusive consumption of one of these products in the total diet and adding the proteins provided by 150 g of meat (chicken fillets were selected as a reference food with 21.8 g of protein/100 g of chicken [24]).

A standard adult between 20 and 39 years old, weighing 70 kg for men and 55 kg for women, was established to select the recommended intake reference. With data provided by Ortega et al., 2019 [21], the RDI protein intake for the standard man was 54 g and 41 g for women. Considering the EFSA guidelines, the PRI was 0.83 g/kg bw daily [23].

In this article, we refer to a high-protein processed food. Adding a nutrient to food is typically defined as fortifying or enriching it. Fortification consists of adding nutrients to foods, whether they already contain them naturally or not, and using foods as vehicles to increase the intake of one or several nutrients in the population. Then fortification can be useful to reduce deficiency problems, while enriched food is the addition of a nutrient or component not initially contained in the food or that has been totally or partially lost in a technological process. In this article, the term high-protein processed food has been used since the products evaluated cannot be considered fortified since they do not respond to a deficiency problem (the opposite situation occurs in Spain and other countries), nor to food enriched since many of the foods to which protein is added already contain it and, in some cases, such as dairy products, are a source of it.

A convenience sample survey of subjects over 18 years of age of both genders (Appendix A) was conducted to accomplish the second part of the study, which was to understand the consumption patterns of these products and assess the surveyed knowledge and general opinions. The survey comprised 15 questions: Three questions collected information regarding the consumer profile, including gender, age, and physical activity. Four questions were aimed at discovering consumers’ knowledge regarding protein-enriched products. Four questions were related to the consumption of protein-enriched products and the reasons and frequency of consumption. Four questions evaluated the perception regarding its usefulness and need for consumption. All questions were formulated in closed form. A Google form was designed for the construction of the questionnaire, and it was spread on social media apps such as WhatsApp, Instagram, Twitter, and Facebook. The study was explained to consumers through an online questionnaire. They were informed that they would participate in the survey using their smartphones and that all data would be de-identified and only reported in the aggregate. Researchers had to obtain a statement of consent from the respondents to participate in the online research by clicking on a statement that they had read and agreed to the terms and conditions. Respondents could only continue with the survey if they stated that they did consent or that they had read the terms and conditions. In order to guarantee their agreement, a screening function was used to direct participants away from the survey if they stated that they did not consent or that they had not read and agreed to the terms and conditions.

A statistical descriptive analysis was carried out on the percentage covered by the recommendation in each category of products: dairy, jellies, energy bars, snacks, breakfast cereals, breads, and creams, which was expressed as a mean and standard deviation. Categorical variables from the online survey data were expressed in absolute and relative frequencies.

## 3. Results

Information on a total of 36 high-protein processed products was compiled. It includes dairy products (yoghurts, milkshakes, drinkable yoghurt, ice creams, mousses, milk, custards, and puddings), jellies, energy bars, snacks (chips, muffins, and cookies), breakfast cereal (muesli), bread, wheat tortillas, pizza dough, and creams (Table 1).

The protein content in these foods varied from 13 to 50 g/100 g of product and between 3.1 and 93 g/serving, while other similar non-enriched choices were between 0 and 11 g/100 g. The size of the offered servings was between 15 and 330 g/serving.

Regarding the current legislation on nutritional and health claims [25], the evaluated products mostly (86%) belonged to foods with a “high protein content” (>20%). The range of protein was from 12 to 88%.

In addition to these statements, practically every product contained other statements such as “0% fat”, “gluten-free”, “no added sugar”, “source of fibre”, and “low on carbohydrates” (Table 2). Table 2 also shows the products targeted at a particular population. It can be observed that such an indication was not present in almost half of the evaluated products.

Concerning the contribution of RDI or PRI [21,22,23], a single serving of enriched products covered 27.6 ± 30.0% of the RDI in women and 21.0 ± 22.8% in men without considering the contribution of any other diet component. According to EFSA references, 34.3 ± 32.3% and 27.0 ± 25.4% were covered for women and men, respectively. In order to have a more comprehensive view of the protein intake, the analysis was recalculated, adding the protein provided by 150 g of meat. In this case, 112.0 ± 18.6% and 84.0 ± 0.14.4% of the RDIs were covered by women and men, respectively. It is essential to highlight that the calculation did not consider the rest of the foods containing protein eaten generally during the day, such as cereals, legumes, eggs, and milk. The contribution of RDI that each one of the products covered with the addition of 150 g of meat was within a range of 87.4% to 306.6% in women and 66.4% to 232.8% in men. Figure 1 and Figure 2 show the percentage covered for each evaluated product grouped into the seven categories for women and men.

Looking at the protein sources of the products analyzed (Table 2), we can observe that whey was the most commonly used type of protein. It is not only the main ingredient in dairy products but also in energy bars, snacks, muesli, and creams. As for jellies, hydrolyzed collagen protein was used, and in snacks such as chips, a soy protein concentrate was added. For bread and cookies, the most commonly added protein was gluten.

Regarding the survey designed to analyze consumption patterns and their knowledge about protein-enriched products, it was completed by 139 participants. A total of 67.6% of the surveyed people affirmed that they had consumed these products. In terms of frequency, 41% stated that they consume them when they feel like it without following a consumption control, 36.7% consumed them because they considered them healthy, 20.9% stated that they consumed them to improve their physical performance, and 12.2% said that they introduced them to their diet to “shape their body”; only 10.8% of the buyers expressed that they purchased these products by direct recommendation from a professional with experience in the topic.

About how they knew these products, 37.4% responded that they saw them in the supermarket and decided to try them, 36.7% said they learned about them from TV or social media advertisements, and 33.8% said they heard about them from friends or family. Only 21.6% stated that they became aware of them based on recommendations from a healthcare professional (nutritionist, pharmacist, nurse, or doctor).

The most popular products chosen were dairy, energy bars, and spreads, which were consumed by 74%, 58.3%, and 25% of respondents, respectively. Despite the frequency of the high-protein products consumption, 80.6% of the participants considered that excessive consumption of them could negatively impact their health.

## 4. Discussion

The results obtained after evaluating a sample of HPPFs widely distributed across supermarkets and online stores allowed us to identify a significant risk to the balance of the diet and health of consumers.

In nutrition, the theoretical relations between diet and health can be represented in a U-shaped graph, where intakes between the minimum requirement and the upper limit are associated with good health. If the nutrient intake is not within those limits, either over or under, it is related to diseases due to excess or deficiency, respectively [12]. The RDI and PRI references guide professionals on the amount of nutrients needed to maintain health in a group of people. Policymakers use them to issue recommendations on nutrient intake to consumers and as the basis for establishing dietary guidelines. These quantitative reference values for nutrient intakes are based on health criteria and range from preventing clinical deficiency to optimizing body stores or status based on scientific evidence. The goal of the RDI and PRI references is to have a low probability of inadequacy while minimizing the potential risk of excess for each nutrient [23].

The analysis of the products offered in this study highlights how they contribute to increasing protein intake in the diet, exceeding the recommendations made by experts. The analysis of the 36 products showed that when consuming one serving, about 50% of the RDI was covered, and the total intake was exceeded by simply considering 150 g of meat in the diet without including any of the remaining protein foods consumed daily. While protein is an essential nutrient for bodily functions, overconsumption poses significant risks. The effect of consuming high amounts of protein in the diet has been widely studied, although the results continue to be controversial or inconsistent and vary according to the type of studies analyzed and the outcome variable analyzed. Recently, an umbrella review of systematic reviews and meta-analyses of observational studies was conducted to evaluate the existing evidence between the intake of dietary proteins and multiple health outcomes. This meta-analysis evaluates twenty unique outcomes and four mortality outcomes (such as all-cause mortality, different kinds of cancer, and coronary heart disease). The authors concluded that dietary protein intake was associated with a higher risk of type 2 diabetes, but the strength of the evidence for the rest of the outcomes was limited [26]. Other reviews have shown that higher total protein intake is associated with higher all-cause mortality, giving special negative attention to protein from meat and dairy [27].

Regarding randomized clinical trials, some systematic reviews and meta-analyses concluded that higher-protein diets probably improve adiposity, blood pressure, and triglyceride levels, but these effects are minor and need to be weighed against the potential for harm [28,29]. However, clinical trials only allow for short- and medium-term evaluation of the effect. In general, the evidence to date forces us to be cautious since the negative effects of high protein consumption cannot be ruled out.

Of course, we must consider special situations or groups that could require or benefit from an increase in protein intake, such as elders with a reduced protein intake or increased needs [4,5]. Vegetarians or people who engage in sports during periods of rapid protein turnover could also be included in this group. It could also be consistent with some people following diets oriented toward weight loss to promote satiety and maintain muscle mass [8]. However, these particular cases cannot be identified if these products are sold in large stores alongside the rest of the food, especially considering that in most of the evaluated products, it was not indicated that they were targeted at a group with specific needs.

Even within the sports field, evidence shows that multiple athletes surpass the guidance values when consuming protein supplements, reaching over three times the recommended intake [10,30,31]. It is not surprising that it is precisely protein in different formats that predominates in sales worldwide [20].

The risk is heightened in cases where food consumption patterns already tend toward a high protein intake, as is the case with Spain [1,16,17] and other countries such as the USA or France [10]. The general perception is that increasing protein intake improves muscle mass and function and contributes to weight control. However, the relationship between protein intake and the postprandial muscle protein synthesis rate is saturable. Likewise, the change in body composition depends on exercise training and the total calorie intake of the diet. Data from both population and randomized controlled studies do not support a clinically meaningful beneficial effect of high protein intake (more than the RDI) on muscle strength and overall physical function. Moreover, studies have shown that high protein intake did not prevent or blunt the age-associated decline in muscle strength, assessed as grip strength and overall physical function [8].

The significant number of nutritional claims made by the studied products suggests that these statements have become more of a marketing strategy than a path to improving people’s health and choices.

In addition, it is essential to consider the type of proteins used in the protein-enriched products to evaluate their nutritional quality since these are determining factors in the metabolic response generated in the body after ingestion. According to this study, 24 of the 36 products analyzed were enriched with dairy whey protein. Whey protein benefits are well described in the literature [32,33]. However, there are few studies investigating the potential adverse effects of a diet with indiscriminate use of this supplement, especially for people with kidney or liver damage or an imbalance in nutrient intake. Overconsumption of whey protein may also contribute to excess animal protein in the diet [34], which is associated with negative outcomes [26,27]. In any case, enrichment of these products with proteins of plant origin, even though it is not necessary on a massive scale, could make more sense due to its greater relationship with positive effects on health [26,27].

The results derived from the survey highlight the popularity of HPPFs consumed by almost three-fourths of the surveyed people. People declared that they did not supervise the frequency at which they were consumed. They declared that they consume these products to increase muscle mass, their performance, or just because they consider them “healthy”. Nevertheless, this is against what scientific evidence reveals because the analysis of the impact of these products on a diet shows how this practice can lead to high protein intakes, further increasing the risk of developing the adverse effects associated with the excessive consumption of this macronutrient [8,10,11]. While 80.6% of the participants expressed awareness that excessive consumption of these types of products could have a negative impact on their health, this should be interpreted with caution, as this response could have been inferred within the context and purpose of the questionnaire.

Although there is a large number of studies that address the effects of a high protein intake on health, to date, we have not identified other studies that evaluate the growth in the supply of these products in the market and analyze their impact on diet and the potential nutritional and health repercussions, especially in populations that, as a consequence of their nutritional habits and culture, have a high protein intake. Nevertheless, other authors, such as Mittendorfer et al. (2020), highlight the risks of the growing demand for protein-fortified food products and their potential adverse public health consequences [8]. It is also necessary to develop clinical trials to quantify the direct clinical and health consequences of this massive supplementation.

As for the limitations of the study, we have to mention the total number of products included. The offer and variety of these products expand daily, so the results of the percentage of the diet covered may be underestimated. It should also be noted that a calculation has been made considering only the consumption of one serving of meat to evaluate the weight of the recommendation. In this case, the data may also underestimate the extent of the problem by not considering other diet components (milk, cheeses, legumes, etc.). However, this shows that the magnitude of the problem may be even greater than what is revealed in this analysis.

Another limitation of this work is that a convenience sample was used for the survey. Convenience sampling is not representative of the population, so the statistical results are not precise, and it is impossible to reach generalized or categorical conclusions, so they should be taken cautiously.

## 5. Conclusions

Considering the findings of the study, it is relevant to warn consumers about the effect of the overconsumption of these high-protein-enriched foods and ensure that they make informed choices, not those driven by advertising or marketing. Likewise, there is a need for regulation by authorities regarding their mass sale or the necessity to include information about who could consume them and the recommended frequencies to do it. Health professionals and nutritional advisors must be permanently informed of the new products offered on the market and offer their patients/clients personalized and adapted consumption recommendations to ensure the balance of their diets. Finally, it is worth noting the importance of nutrition education for the population to make healthy choices.

## Figures and Tables

**Figure 1 nutrients-16-01697-f001:**
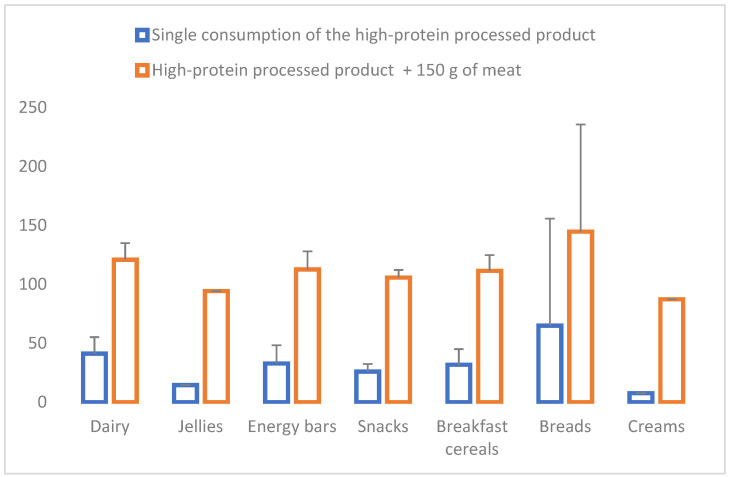
Contribution to the Recommended Daily Intakes of proteins in women (%) (references [21,22]) considering the single consumption of the high-protein processed product and this product + 150 g of meat (mean and standard deviation of products grouped into seven categories: dairy, jellies, energy bars, snacks, breakfast cereals, breads, and creams).

**Figure 2 nutrients-16-01697-f002:**
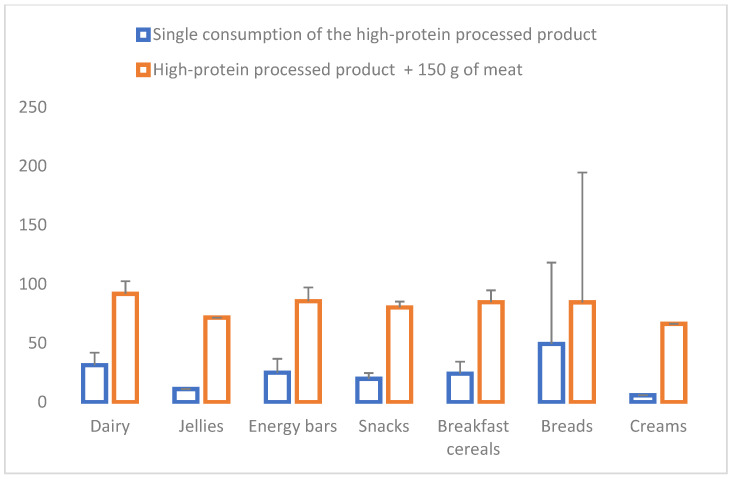
Contribution to the Recommended Daily Intakes of proteins in men (%) (references [21,22]) considering the single consumption of the high-protein processed product and this product + 150 g of meat (mean and standard deviation of products grouped into seven categories: dairy, jellies, energy bars, snacks, breakfast cereals, breads, and creams).

**Table 1 nutrients-16-01697-t001:** Characteristics of the evaluated products concerning their nutritional composition and the contribution to the recommended intakes of proteins.

Kind of Product	Brand	Proteins (g/100 g EP)	Proteins (g/100 g NEP)	Container/Serving Size (g or mL)	Proteins (g/Container or Serving)	Energy/Container or Serving (kcal)	Energy (from Proteins)/Container or Serving (kcal)	Energy (from Proteins)/Container or Serving (%)	RDI Proteins/Container or Serving (%) ^3^		PRI Proteins/Container or Serving (%) ^4^	
									Woman	Man	Woman	Man
Dairy												
Yoghurts	LIDL ^1^	10.0	3.3	200.0	20.0	130.0	80.0	61.5	48.8	37.0	43.8	34.4
	Mercadona ^2^	8.3		120.0	10.0	64.0	40.0	62.5	24.4	18.5	21.9	17.2
	Yopro	9.4		160.0	15.0	88.0	60.0	68.2	36.6	27.8	32.9	25.8
Milkshakes	LIDL ^1^	10.5	3.9	166.0	17.4	109.0	69.6	63.8	42.4	32.2	38.1	29.9
	Mercadona ^2^	7.9		330.0	26.0	151.0	104.0	68.9	63.4	48.1	57.0	44.7
	Yopro	6.0		330.0	15.0	118.0	60.0	50.8	36.6	27.8	32.9	25.8
	Prozis	6.0		250.0	15.0	90.0	60.0	66.7	36.6	27.8	32.9	25.8
Drinkable yoghurt	Mercadona ^2^	7.1	3.0	280.0	20.0	143.0	80.0	55.9	48.8	37.0	43.8	34.4
	Yopro	8.3		300.0	25.0	177.0	100.0	56.5	61.0	46.3	54.8	43.0
Ice creams	LIDL ^1^	7.6	3.5	45.5	3.5	116.0	14.0	12.1	8.5	6.5	7.7	6.0
	Mercadona ^2^	9.3		265.0	24.6	389.0	98.4	25.3	60.0	45.6	53.9	42.3
Mousses	Mercadona ^2^	10.0	4.1	200.0	20.0	152.0	80.0	52.6	48.8	37.0	43.8	34.4
	Yopro	10.1		200.0	20.0	155.0	80.0	51.6	48.8	37.0	43.8	34.4
	Valio profeel	12.0		150.0	18.0	130.5	72.0	55.2	43.9	33.3	39.4	31.0
	Reina	10.0		100.0	10.0	123.0	40.0	32.5	24.4	18.5	21.9	17.2
Milk	Mercadona ^2^	6. 0	3.9	250.0	15.0	142.0	60.0	42.2	36.6	27.8	32.9	25.8
Custards	Mercadona ^2^	10.0	5.0	120.0	12.0	92.0	48.0	52.2	29.3	22.2	26.3	20.6
Pudding	Yopro	10.0	3.2	180.0	18.0	82.0	72.0	87.8	43.9	33.3	39.4	31.0
Jellies												
Jelly	Carrefour	6.0	0.0	100.0	6.00	30.0	24.0	80.0	14.6	11.1	13.1	10.3
	Mercadona	6.0		100.0	6.00	39.0	24.0	61.5	14.6	11.1	13.1	10.3
Energy bars												
Bar	LIDL	50.0	5.7	45.0	22.5	164.0	90.0	54.9	54.9	41.7	49.3	38.7
	Prozis	30.0		35.0	10.5	140.0	42.0	30.0	25.6	19.4	23.0	18.1
	Foodspring	29.0		45.0	13.0	157.0	52.0	33.1	31.7	24.1	28.5	22.4
	El almendro	24.0		35.0	8.0	180.0	32.0	17.8	19.5	14.8	17.5	13.8
Snacks												
Chips	PROZIS	45.0	6.5	25.0	11.2	100.7	45.0	44.7	27.4	20.8	24.64	19.36
Muffin	Prozis	13.0	3.5	60.00	7.80	175.2	31.2	17.8	19.0	14.4	17.1	13.4
Cookie	Foodspring	26.0	6.0	50.0	13.0	227.0	52.0	22.9	31.7	24.1	28.5	22.4
Breakfast cereal												
Muesli	Prozis	23.0	7.0	40.0	9.2	159.4	36.8	23.1	22.4	17.0	20.1	15.8
	Foodspring	28.2	7.0	60.0	16.9	263.0	67.6	25.7	41.2	31.3	37.0	29.1
Breads												
Bread	Prozis	17.0	11.0	30.0	5.1	73.5	20.4	27.7	12.4	9.4	11.2	8.8
	Keto protein	27.0		50.0	13.5	116.0	54.0	46.5	32.9	25.0	29.6	23.2
Toasts	Mercadona	46.5	10.0	200.0	93.0	818.0	186.0	22.7	226.8	172.2	203.7	160.1
Wheat tortillas	Keto protein	22.0	7.0	40.0	8.8	127.0	35.2	27.7	21.5	16.3	19.3	15.1
Pizza dough	Keto protein	28.0	6.50	45.0	13.0	110.0	52.0	47.3	31.7	24.1	28.5	22.4
Creams												
Cocoa cream	Prozis	21.0	6.3	15.0	3.1	77.4	12.6	16.3	7.7	5.8	6.9	5.4
Hazelnut cream	Foodspring	21.0	6.3	15.0	3.1	79.5	12.6	15.8	7.7	5.8	6.9	5.4

EP: enriched product; NEP: not-enriched product; ^1^ LIDL (type “high protein”); ^2^ Mercadona (type “+ proteins”); ^3^ RDI: Recommended Daily Intakes, reference: [21,22]; ^4^ PRI: population reference intake, reference: [23].

**Table 2 nutrients-16-01697-t002:** Characteristics of the evaluated products regarding the type of added protein, claims, and the presence or absence of the indication of the person to whom the product is targeted.

Kind of Product	Brand	Type of Added Protein	Claim/Statement	Indication of Whom the Product Is Intended for (Yes/No)
Dairy				
Yoghurts	LIDL ^1^	DWP	Lactose-free and low in fat	NO
	Mercadona ^2^	DWP	0% fat, gluten free	NO
	Yopro	DWP	0% fat, 0% added sugar, without artificial colourings or preservatives	Athletes, it indicates that it is ideal to enhance training
Milkshakes	LIDL ^1^	DWP	Gluten-free, no added sugar, and low in fat	NO
	Mercadona ^2^	DWP	Source of vitamin B_6_, lactose-free, and without added sugars	NO
	Yopro	DWP	0% fat, 0% added sugar, without colourings or preservatives	Athletes, it indicates that it is ideal to enhance training
	Prozis	DWP	Low fat	Children, seniors, athletes, busy professionals, and weight loss programs
Drinkable yoghurt o	Mercadona ^2^	DWP	0% fat, gluten free	NO
	Yopro	DWP	0% fat, 0% added sugar, without artificial colourings or preservatives, without lactose	Athletes, it indicates that it is ideal to enhance training
Ice creams	LIDL ^1^	DWP	No added sugars	NO
	Mercadona ^2^	DWP	NO	NO
Mousses	Mercadona ^2^	DWP, animal jelly	Lactose-free	NO
	Yopro	DWP, animal jelly	Low fat, 0% added sugars	Athletes, it indicates that it is ideal to enhance training
	Valio profeel	DWP	Rich in protein, added sugars free and lactose-free	NO
	Reina	DWP	High in protein, low fat, 0% added sugar	NO
Milk	Mercadona ^2^	DWP	Lactose-free. Enriched with protein and calcium	NO
Custards	Mercadona ^2^	DWP	Gluten-free. Source of protein and calcium	NO
Pudding	Yopro	DWP	Lactose-free and low in fat	NO
Jellies				
Jelly	Carrefour	Hydrolyzed collagen protein, jelly	High protein content, 0% fat, gluten free	NO
	Mercadona	Hydrolyzed collagen protein, jelly	Gluten free, 0% fat	NO
Energy bars				
Bar	LIDL	Mix of proteins ^1^	50% proteins	NO
	Prozis	DWP	30% proteins, source of fibre, GMO-free	People who control their diet, athletes, busy people
	Foodspring	DWP	No added sugars	Athletes
	El almendro	Pea protein extruded ^2^	Source of protein, with almonds, source of fibre, gluten-free, palm oil-free	NO
Snacks				
Chips	Prozis	Soy protein concentrate	High protein content (45%), no added sugars, high fiber content	Athletes, busy people, and people who control their diet
Muffin	Prozis	DWP	Low in sugar, source of protein, without aspartame, without artificial colourings	Suitable for all persons
Cookie	Foodspring	Wheat protein (gluten)	Low sugar, high protein	NO
Breakfast cereal				
Muesli	Prozis	DWP	High protein content and high fiber content	NO
	Foodspring	Soybean flakes, almond flakes, extruded soybeans, cashews, toasted hazelnuts, protein sunflower seeds (sunflower protein, rice flour)	High protein content, 100% organic, GMO-free, rich in fibre	NO
Breads				
Bread	Prozis	Whole wheat flour (gluten), wheat flour, wheat gluten, wheat bran, soybean flakes, barley malt flour	High protein and low carbohydrate	Athletes and weight control
	Keto protein	Whole wheat flour (gluten), wheat flour, wheat gluten, wheat bran, soybean flakes, barley malt flour	Low in carbohydrates, rich in protein, fibre contribution	Healthy lifestyle, weight control or weight maintenance, sport/bodybuilding
Toasts	Mercadona	Vegetable flours (rice protein, whole rye flour (gluten), whole chickpea flour, hydrolyzed wheat protein (gluten)	Low carb	NO
Wheat tortillas	Keto protein	Wheat protein, pea protein, rice protein	Low in carbohydrates, high in protein, low in sugar and high in fibre	Healthy lifestyle, weight control or weight maintenance, sport/bodybuilding
Pizza dough	Keto protein	Wheat protein, sunflower seed meal, soy protein	Low in carbohydrates, high in protein, low in sugar and high in fibre	Healthy lifestyle, weight control or weight maintenance, sport/bodybuilding
Creams				
Cocoa cream	Prozis	DWP	No palm oil, reduced salt content, no added sugar, no aspartame, no artificial colourings or preservatives, GMO-free	NO
Hazelnut cream	Foodspring	DWP	No added sugar, no palm oil, high protein, and low carbohydrate	NO

DWP = dairy whey protein; ^1^ mix of proteins (includes collagen hydrolyzate, milk protein, soy protein isolate, whey protein concentrate, whey protein isolate); ^2^ Pea Protein Extrudate (includes Pea Protein Isolate, Pea Protein Concentrate, Tapioca Starch); GMO = genetically modified organisms.

## Data Availability

The original contributions presented in the study are included in the article, further inquiries can be directed to the corresponding author.

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
