# Peer review of "High-Protein Processed Foods: Impact on Diet, Nutritional Status, and Possible Effects on Health"

_nutrients, 2024, doi:10.3390/nu16111697_

Round 1

Reviewer 1 Report (Previous Reviewer 1)

Comments and Suggestions for Authors

The authors improved the paper by fixing the critical points I had pointed out in the previous review.

I think the paper is now ready for publication. 

Author Response

Thank you for your comments.

Yours sincerely,

Viviana Loria Kohen, PhD, MSc

Department of Nutrition and Food Science.

Faculty of Pharmacy.

Complutense University of Madrid

[email protected] +34 91 394 1809

Reviewer 2 Report (New Reviewer)

Comments and Suggestions for Authors

Ortega and colleagues present an interesting article of the use of high protein processed foods (HPPF) and its possible effects.

The authors addressed that the high use of proteins in food may cause problems. This is still a controversial subject that is not fully understood, as several of the articles mentioned say.

The real innovative subject of this article has to do with the use and abuse of various processed foods containing high doses of protein and their daily accumulation.

The main author uses a lot his previous articles as references and new and interesting literature should also be included, in order to enrich the work.

I beleive that the impact of these HPPF in food can not be made with this superficial questionnaire and so, this part of the work cant be part of the title or it will engage the reader, since these are only assumptions and not conclusions of this work.

So, I believe that the title should be changed to reflect the real work to: High-protein processed foods: impact on diet, nutritional status and possible repercussions or possible effects on health

Author Response

Thank you very much for taking the time to review this manuscript. Please find the detailed  responses below and the corresponding revisions highlighted/in track changes in  the re-submitted files.

This manuscript is a resubmission of an earlier submission. The following is a list of the peer review reports and author responses from that submission.

Round 1

Reviewer 1 Report

Comments and Suggestions for Authors

The paper 'High-protein processed foods: impact on diet nutritional status and health' examines the impact of high-protein processed foods on diet and health.

Several aspects of the paper, including methods and conclusions, need substantial improvement.

1. Deepen statistical analysis: Use more advanced statistical methods for a more robust analysis of data.

2. Comparing plant and animal proteins: Analysing the differences between plant and animal protein products in terms of their impact on diet and health.

3. Detail data collection methods: Provide more detail on data collection and selection methods to increase the transparency of the study.

4. Deepen the analysis of the results: Expand the discussion of the results, considering the implications for public health and dietary recommendations.

5. Include a broader literature review: Expand the existing literature review to provide a more complete context to the results.

Reviewer 2 Report

Comments and Suggestions for Authors

 This stusy sought to analyze a sample of high-protein processed foods (HPPF) available in different supermarkets and evaluate how their consumption can increase excessively the percentage covered of the recommended intake of that nutrient. Moreover, the other objective was to evaluate the  rate of people who buy these products and the reasoning behind their choices. Although this is a great study, i have several changes:

-This study is more adequate for food journal.

-Introduction:

What is hypothesis?

-Methods:

Local? How was choice of foods? 

-Results:

Too much descriptive. How body composition of consumers?

-Discussion:

Lack a deep discussion regarding to body composition and type the food.

Round 2

Reviewer 1 Report

Comments and Suggestions for Authors

The authors have made some changes to the paper.

Critical issues still remain:

Clarification and expansion of research objectives: The objectives could be defined more explicitly to highlight the novel aspects of the study and how it advances existing knowledge on high-protein processed foods (HPPFs).

Methodological detail: More details on the selection criteria for HPPFs, the survey design and the statistical methods used for data analysis could improve the reproducibility of the study.

Discussion of limitations: A more in-depth discussion of the limitations of the study, including the convenience sample used for the survey and how this may affect the generalisability of the results.

Comparison with previous studies: A more in-depth comparison with existing literature to contextualise the findings in the broader research landscape would strengthen the paper's contribution to the field.

Implications and recommendations: Broaden the practical implications of the findings for consumers, policy makers and practitioners and provide more concrete recommendations based on the findings of the study.

Reviewer 2 Report

Comments and Suggestions for Authors

Again: this study is more adequate for food journal
